# Electronic medical record implementation in the emergency department in a low-resource country: Lessons learned

Nagham Faris[1], Miriam Saliba[1], Hani Tamim[2,3], Rima Jabbour[1], Ahmad Fakih[4], Zouhair Sadek[4], Rula Antoun[4], Mazen El Sayed[1]*, Eveline Hitti[1]*

1 Emergency Department, American University of Beirut Medical Center, Beirut, Lebanon, 2 Department of Internal Medicine, American University of Beirut Medical Center, Beirut, Lebanon, 3 College of Medicine, Alfaisal University, Riyadh, Saudi Arabia, 4 Department of Information Technology, American University of Beirut Medical Center, Beirut, Lebanon

* eh16@aub.edu.lb (EH); melsayed@aub.edu.lb (MES)

## Abstract

### Objective

There is paucity of information regarding electronic medical record (EMR) implementation in emergency departments in countries outside the United States especially in low-resource settings. The objective of this study is to describe strategies for a successful implementation of an EMR in the emergency department and to examine the impact of this implementation on the department's operations and patient-related metrics.

### Methods

We performed an observational retrospective study at the emergency department of a tertiary care center in Beirut, Lebanon. We assessed the effect of EMR implementation by tracking emergency departments' quality metrics during a one-year baseline period and one year after implementation. End-user satisfaction and patient satisfaction were also assessed.

### Results

Our evaluation of the implementation of EMR in a low resource setting showed a transient increase in LOS and visit-to-admission decision, however this returned to baseline after around 6 months. The bounce-back rate also increased. End-users were satisfied with the new EMR and patient satisfaction did not show a significant change.

### Conclusions

Lessons learned from this successful EMR implementation include a mix of strategies recommended by the EMR vendor as well as specific strategies used at our institution. These can be used in future implementation projects in low-resource settings to avoid disruption of workflows.

**Data Availability Statement:** All relevant data are within the paper.

**Funding:** The author(s) received no specific funding for this work.

**Competing interests:** The authors have declared that no competing interests exist.

# Introduction

Electronic Medical Records (EMRs) adoption in hospitals and emergency departments (EDs) is increasing, especially in high-resource countries where healthcare digitization is often incentivized by governments and policymakers [1, 2].

The adoption of these EMRs in hospitals often entails challenges including high cost, end-users resistance, and complex implementation processes [3]. A failed EMR implementation incurs significant financial losses to healthcare organizations [4]. An extensive preparedness plan is therefore required for a successful implementation especially in the busy and complex ED setting [5].

EMRs have been shown to improve clinical practice and patient safety in hospitals and EDs, where timely clinical information is needed to manage patients with life threatening emergency medical conditions [1]. Additional benefits include a decrease in medical errors [6–8] and unnecessary testing [9], reduction in length of stay (LOS) [10–12], and improved documentation with less illegible, misplaced, or lost documents [5].

There is paucity of information regarding EMR implementation in EDs in countries outside the United States, especially in low-resource settings. The objective of this study is to describe strategies for a successful implementation of an EMR in the ED and examine the impact of this implementation on this department's operations and patient-related metrics at a tertiary care center in Beirut, Lebanon.

# Methods

## Study setting and design

This observational retrospective study took place at the largest academic tertiary care center in Lebanon, in an ED that is divided into three sections (High Acuity, Low Acuity, and Pediatrics), with over 55,000 patient visits per year. The ED has an associated residency program in emergency medicine and is staffed by a mix of American board-certified/eligible Emergency Medicine physicians as well as non-Emergency Medicine physicians with extensive experience in emergency care. The study was approved by the institutional review board of the American University of Beirut, and requirement for informed consent was waived. De-identified data regarding all variables was obtained from ED leadership.

This EMR implementation was the first of its kind in Lebanon, a low-middle income country. In this low-resource setting, financial resources were limited. There is an absence of government funding for healthcare infrastructure, and healthcare institutions rely on internal funding. Moreover, there was a lack of trained personnel to implement and maintain the EMR. Budget to hire external consultants and to pay faculty and staff additional compensation for roles performed during Go-Live was limited.

Before implementation of the EMR, an electronic dashboard provided real-time information about active ED patients. This dashboard was used to access the patient's records through a homegrown HIS (Health Information System). It was also used for the CPOE (Computerized Physician Order Entry), admission orders, charges, and discharge orders. Laboratory and radiology orders were electronically placed but some orders still required paper forms (including medications, cytology, and echocardiography). Paper-based documents were used for triage, physician notes, nursing notes, and orders. The paper charts were then scanned by the medical records department and uploaded to the homegrown HIS that displays previous patient encounters and diagnostic test results. Different applications were also present including Admission, Discharge and Transfer (ADT), registration and Master Patient Index, material management, and billing. Lab information system and PACS had also been installed in prior

years. For pharmacy, a plan was to install an automated medication dispensing system (BD Pyxis™ MedStation™ ES) during the implementation of the new EMR. The projected implementation plan of the EMR (EPIC Systems Corporation) was over 2 years, which was proposed and agreed on by the EMR vendor. For ED staff, the plan consisted of extensive workflow analysis sessions, adoption sessions, training, and preparation. On November 3, 2018, the new EHR system (EPIC) went live across all clinical areas including the ED.

## Emergency department and organizational strategies

**Pre-implementation: ED leadership engagement and operational strategies.** ED leadership was engaged fully in all implementation phases. Extensive sessions over several months were dedicated to workflow analysis, ED processes, staffing and hardware requirements, and issues with interfaces. All clinical and non-clinical risks were reviewed, and corresponding mitigation plans were prepared. Paper-based order sets and protocols were modified and transferred to the EMR.

**Pre-implementation: Staff training and engagement strategies.** All ED physicians, nurses, and staff underwent extensive training that included: online training, hands-on training, and practice through the EMR 'playground'. Prerequisite online introductory videos were sent to end-users through the institution's online platform for easy access and tracking. End-users then registered for small group training sessions and were able to choose times and dates that suit their schedules.

Each physician attended a four-hour class in which the basic EMR functions were presented, another two-hour class in which personalization of physician notes and orders were introduced, and a third two-hour refresher course. Also, physicians were divided in groups, each led by one physician super-user, to practice personalization functions and tips. This occurred over a period of two months.

In our ED, a physician champion was chosen to act as the link between the ED leadership, ED physicians, and Information Technology (IT) staff. Super-users were also selected from the ED. These included physicians, residents, medical students, and nurses who underwent advanced training which included troubleshooting strategies that could be used during and after Go-Live. Each super-user was assigned a group of 4 to 5 trainees for one-on-one support as needed.

Physicians who needed additional sessions, including those who were less computer savvy, had visual problems, or had infrequent ED shifts, were identified in advance and received additional personalized training from super-user physicians.

With the involvement of all stakeholders, several simulations and drills took place reflecting the new EMR workflows, with special focus on high-risk areas. All possible technical glitches were addressed and minimized prior to Go-Live. Implementation teams from the EMR vendor were also available on-site. A cut-over team was created and included a physician, two nurses, patient access staff, cashiers and IT staff. A cut-over drill took place, problems were identified, and the drill was repeated three times until all problems were solved and the process occurred smoothly.

Moreover, multiple new computers with wide screens were installed. A workstation was allocated for every person working in the ED. Staff were instructed to use their designated workstation even before Go-Live. Workstations on wheels (WOWs) and wall-mounted stations were also installed. New printers were installed to ensure adequate printing capabilities.

**Post-implementation: EMR Go-Live strategies.** Go-Live occurred on November 3, 2018, which was a Saturday. Patient load in the hospital during the weekend is usually lower than weekdays because of clinic closure and lack of scheduling of elective cases. Adequate IT

support was available for the ED. The cut-over team was present during Go-Live. Physician and nurses involved in cut-over did not have clinical responsibilities and focused on ensuring a smooth cut-over process.

Super-users had no clinical responsibilities during super-user shifts and were available at all times for the first three days of Go-Live. In addition, the ED attending staffing model included at least one physician at all times who had also undergone super-user training. This staffing model was maintained for two weeks post implementation. Super-user coverage was modified according to the need. Staffing was also increased with an extra attending present in the high turnover low-acuity area of our ED during the first 3 weeks after implementation.

A structured troubleshooting system was created. All super-users were also trained on how to place tickets for issues that could not be solved real-time. Tickets were placed online, and the super-users included the priority of the problem (low, medium, high, or critical) as well as important details needed to facilitate troubleshooting including screenshots of alerts, workstations involved, and user information. IT response was effective in addressing and resolving tickets immediately after Go-Live.

Daily huddles post Go-Live took place to discuss issues, updates and plans. A list of tips was provided daily and disseminated by super-users to end-users. Moreover, multidisciplinary teams had regular meetings to discuss issues and propose solutions. A close follow-up and collaboration occurred between the ED leadership, ED staff, and the IT staff.

Knowing that operational data and metrics can be inaccurate after Go-Live [13], emphasis was placed on report validation. The top ten key operational metrics were prioritized and report validation started immediately after Go-Live. Observations were made, compared and benchmarked to historical metrics and live observations. This ensured reliable data regarding key ED metrics after Go-Live.

## Outcomes and intervention results

The main outcomes assessed reflected the priority ED operational quality metrics including LOS, door to admission request, and rate of return visits or bounce-backs. The bounce-back rate captures the percentage of ED patients who return to the ED within 72 hours after being discharged from the ED. The high-risk bounce-back rate shows the percentage of patients who return within 72 hours and require admission, are dead, or get transferred to another hospital. Secondary outcomes included end-user satisfaction and patient satisfaction.

## Data collection

Data was collected for the period of one year before EMR implementation (Nov. 2, 2017 until Nov. 2, 2018) and one year after (Nov. 3, 2018 until Nov. 3, 2019). For the pre-intervention period, data regarding ED quality metrics was collected from a business intelligence solution that was adopted as well as from electronic reports reading from e-forms. For the post intervention period, reports were obtained from the new EMR and were validated.

After implementation, three online surveys were used to assess end-users' satisfaction with the newly-implemented EMR. The surveys were sent at one month, three months, and one year after implementation. End-users included physicians, nurses, and other ED staff. End-users' answers were on a scale from 1 to 6 where 1 corresponds to "Strongly Disagree" and 6 corresponds to "Strongly agree". A satisfaction score was created.

A representative from the patient affairs office regularly conducted phone surveys with a random selection of patients who presented to the ED as part of a routine quality initiative. Phone calls took place one to two days after patient discharge. Surveys consisted of 30

questions divided according to 6 domains. These surveys were conducted before and after implementation, except for the first two weeks after implementation.

## Analysis

The Statistical Package for Social Sciences (SPSS), version 24.0 was used for data cleaning, management, and analyses. Categorical variables were summarized using numbers and percentages while continuous variables were summarized by means ± standard deviation (SD). The association between pre-implementation/post-implementation, and other categorical variables was carried out by using the Chi-square test. Student's t-test was used for the association with continuous variables. Moreover, Cochran-Mantel-Haenszel Statistics was used to evaluate the p for trend for categorical variables over time. P-value < 0.05 was used to indicate statistical significance.

## Results

Table 1 shows the baseline and demographic characteristics of the ED patient population 1 year pre- and 1 year post-implementation.

### Operational outcomes

Mean ED LOS increased in the post-implementation period, as shown in Fig 1; however mean LOS returned to its baseline at around 6 months post-implementation. A similar trend was observed for the mean ED visit to admission decision time (Fig 2). Fig 3 presents the bounce-back rate, the results show an increase from 3.4% to 3.7% after implementation. The high-risk bounce-back rate also increased from 1.1% to 1.2%.

**Table 1. Baseline and demographic characteristics.**

|  |  | Pre N = 54893 (%) | Post N = 56417 (%) | p-value |
|---|---|---|---|---|
| **Age (years)** | Mean (±SD) | 35.7 ± 24.8 | 25.3 ± 24.8 | <0.0001 |
| **Gender** | Male | 27456 (50.0) | 28584 (50.7) | 0.03 |
|  | Female | 27437 (50.0) | 27833 (49.3) |  |
| **Country** | Lebanon | 48220 (87.8) | 55250 (97.9) | <0.0001 |
|  | Others | 6673 (12.2) | 1167 (2.1) |  |
| **Guarantor** | Self-payer | 10201 (18.6) | 9331 (16.5) | <0.0001 |
|  | Insured | 44692 (81.4) | 47073 (83.5) |  |
| **ESI** | Mean (±SD) | 3.01 ± 0.35 | 2.95 ± 0.37 | <0.0001 |
|  | 1–2 | 2647 (4.8) | 4929 (8.7) | <0.0001 |
|  | 3 | 49340 (89.9) | 49104 (87.0) |  |
|  | 4–5 | 2906 (5.3) | 2384 (4.2) |  |
| **ED visit to admission decision (minutes)** | Mean (±SD) | 138.80 ± 103.80 | 151.43 ± 111.10 | <0.0001 |
| **ED disposition** | Admitted | 11074 (20.2) | 11698 (20.7) | <0.0001 |
|  | AMA | 2582 (4.7) | 2650 (4.7) |  |
|  | Dead | 94 (0.2) | 105 (0.2) |  |
|  | Home | 40783 (74.3) | 41531 (73.6) |  |
|  | Transfer to other hospital | 360 (0.7) | 433 (0.8) |  |
| **LOS (hours)** | Mean (±SD) | 2.84 ± 3.33 | 3.26 ± 5.25 | <0.0001 |
| **Bounce-back** | Yes | 1873 (3.4) | 2109 (3.7) | 0.003 |
| **High-risk bounce-back** | Yes | 583 (1.1) | 699 (1.2) | 0.01 |

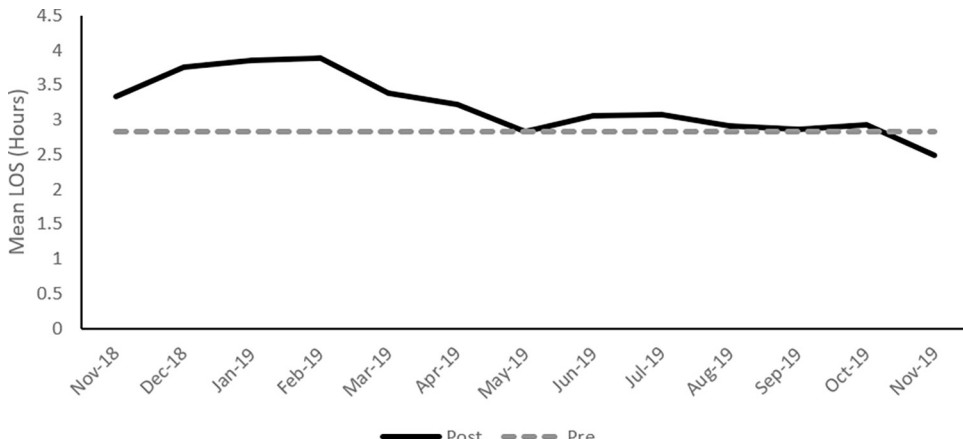

**Fig 1. Mean length of stay (LOS) for the post-implementation period with the mean of the pre-implementation value as baseline.**

## End-user satisfaction

Table 2 shows the result of the three conducted online surveys. End-users were satisfied with the newly implemented EMR. After implementation, and with time, end-users experienced a significant improvement in patient care (4.12 ± 1.49 vs5.14 ± 1.35, p-value = 0.01) and reported a better efficiency in their day-to-day jobs (4.80 ± 1.17 vs 5.47 ± 0.83, p-value = 0.04).

## Patient satisfaction

Table 3 shows the results of the patient satisfaction interviews that were conducted. No significant difference was found in the overall patient satisfaction pre- and post-implementation; only three variables scored significantly higher when comparing post to pre-implementation: the courtesy of the staff in registration (4.60 ± 0.57 vs 4.66 ± 0.57, p-value = 0.02), waiting time at registration area (4.43 ± 0.74 vs 4.58 ± 0.67, p-value <0.0001) and LOS before triage (4.53 ± 0.69 vs 4.63 ± 0.63, p-value = 0.001).

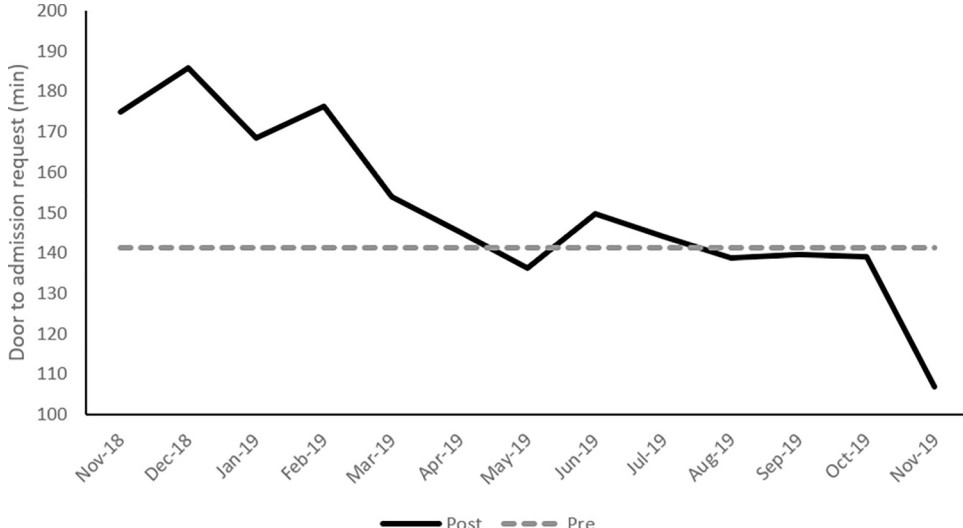

**Fig 2. Mean visit to admission decision time for the post-implementation period by month with the mean of the pre-implementation period as baseline.**

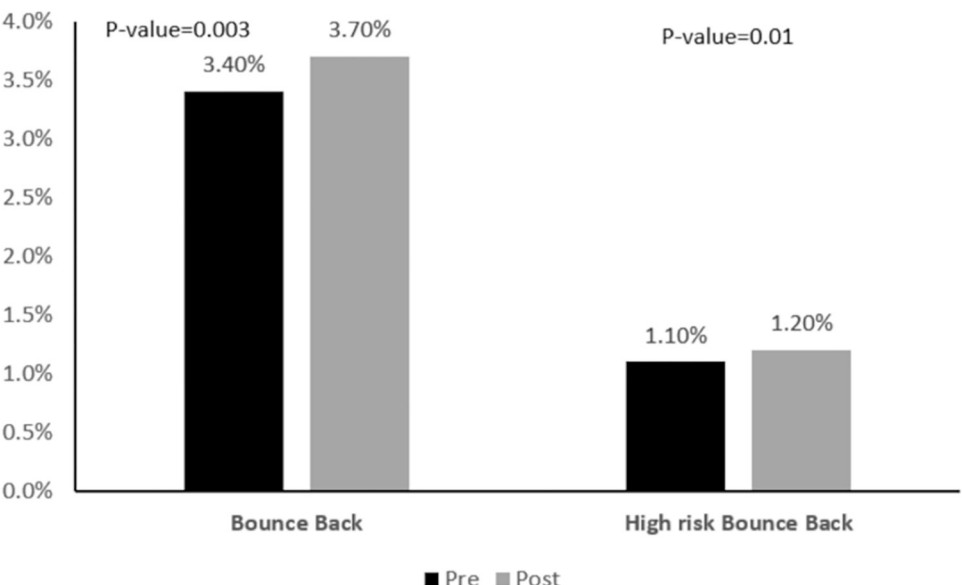

**Fig 3. Bounce-back and high-risk bounce-back rate in the pre- and post-implementation periods.**

## Discussion

In this study, we describe the implementation of HIMSS stage 6 EMR in a setting that used a hybrid model EMR and paper-based charting.

Results presented describe the operational and clinical impact of this EMR implementation. This implementation was considered successful based on vendor and hospital assessment. Main operational metrics including mean patient LOS and mean door to admission decision returned to baseline values after around 6 months, end-user satisfaction in the ED was relatively high, and patient satisfaction was not significantly affected.

Statistically significant differences are seen in the demographic characteristics between pre and post groups. However, most of these differences are mainly due to the large sample size and are not clinically significant. The pre-implementation patient group seems to be significantly older than the post-implementation group, which might seem to bias the operational metrics results such as LOS. For this reason, a month-by-month analysis was done, and the

**Table 2. Results of the surveys assessing end-user satisfaction (on a scale from 1 to 6).** The numbers shown represent the mean.

|  | First survey N = 44 | Second survey N = 15 | Third survey N = 15 | P-value |
|---|---|---|---|---|
| **Adequate training** | 4.68 ± 1.25 | 5.20 ± 1.08 | 5.07 ± 1.39 | 0.31 |
| **Adequate super-user support** | 5.19 ± 1.33 | 5.79 ± 0.43 | 5.08 ± 1.12 | 0.20 |
| **Ease and accuracy of capturing charges** | 4.58 ± 1.55 | 5.20 ± 1.03 | 4.90 ± 0.74 | 0.42 |
| **Ease of EMR use** | 4.95 ± 1.13 | 5.47 ± 0.64 | 5.27 ± 1.03 | 0.21 |
| Better efficiency | 4.80 ± 1.17 | 5.40 ± 0.63 | 5.47 ± 0.83 | **0.04** |
| Improved patient care | 4.12 ± 1.49 | 5.07 ± 0.59 | 5.14 ± 1.35 | **0.01** |
| Improved revenue cycle | 4.60 ± 1.24 | 5.20 ± 0.63 | 5.11 ± 0.60 | 0.20 |
| Satisfactory system response time | 4.52 ± 1.19 | 4.53 ± 1.30 | 4.80 ± 1.01 | 0.73 |
| Feeling supported in EMR use | 4.86 ± 0.98 | 5.00 ± 0.85 | 5.13 ± 1.25 | 0.66 |
| Ability to take appropriate actions from reports | 4.68 ± 1.14 | 5.07 ± 0.80 | 4.53 ± 1.46 | 0.41 |
| Overall Satisfaction | 4.84 ± 1.16 | 5.27 ± 0.70 | 4.93 ± 1.34 | 0.45 |

**Table 3. Comparison of ED patients satisfaction score between pre-implementation and post-implementation (The scale score based on: 1 = very poor; 2 = poor; 3 = fair; 4 = good; 5 = very good).**

| | | Pre-N = 871 | Post-1 year N = 1156 | p-value |
|---|---|---|---|---|
| **Admission** | Courtesy of staff in registration area | 4.60 ± 0.57 | 4.66 ± 0.57 | **0.02** |
| | Explanations given by registration staff | 4.57 ± 0.62 | 4.59 ± 0.59 | 0.44 |
| | Cooperation/helpfulness of registration staff | 4.63 ± 0.56 | 4.68 ± 0.54 | 0.06 |
| | Waiting time at registration area | 4.43 ± 0.74 | 4.58 ± 0.67 | **<0.0001** |
| **Nursing** | LOS before triage | 4.53 ± 0.69 | 4.63 ± 0.63 | **0.001** |
| | Nurses introduced themselves | 4.46 ± 0.81 | 4.50 ± 0.78 | 0.37 |
| | Courtesy of nurses | 4.68 ± 0.57 | 4.70 ± 0.57 | 0.69 |
| | Listening to questions and concerns | 4.69 ± 0.58 | 4.65 ± 0.61 | 0.15 |
| | Communication of progress and delays | 4.37 ± 0.85 | 4.35 ± 0.87 | 0.72 |
| | Explanations given by nurses (e.g.: tests/procedures) | 4.51 ± 0.71 | 4.52 ± 0.68 | 0.88 |
| | Responsiveness of nurses to requests and needs | 4.56 ± 0.66 | 4.50 ± 0.74 | 0.07 |
| | Pain assessment by nurses | 4.54 ± 0.69 | 4.53 ± 0.67 | 0.75 |
| **Medical Team** | LOS before being seen by ED physician | 4.30 ± 0.88 | 4.30 ± 0.90 | 0.86 |
| | Physicians introduced themselves | 4.59 ± 0.68 | 4.60 ± 0.64 | 0.89 |
| | Courtesy of attending physician | 4.77 ± 0.52 | 4.77 ± 0.51 | 0.91 |
| | Explanations physician gave about condition | 4.68 ± 0.60 | 4.67 ± 0.62 | 0.90 |
| | Communication of plan of care | 4.67 ± 0.60 | 4.65 ± 0.62 | 0.43 |
| | Concern the physician showed for questions or worries | 4.68 ± 0.58 | 4.66 ± 0.60 | 0.39 |
| | Instructions physician gave about follow up care | 4.61 ± 0.62 | 4.58 ± 0.67 | 0.23 |
| | Amount of time physician spent with patient | 4.54 ± 0.67 | 4.52 ± 0.65 | 0.50 |
| **Discharge** | Courtesy of cashiers | 4.56 ± 0.65 | 4.57 ± 0.65 | 0.81 |
| | Explanations cashiers gave | 4.55 ± 0.65 | 4.45 ± 0.77 | 0.13 |
| | Cooperation/helpfulness of cashiers | 4.55 ± 0.70 | 4.56 ± 0.64 | 0.83 |
| | Waiting time at cashier's station for payment | 4.50 ± 0.74 | 4.52 ± 0.71 | 0.79 |
| **Overall Satisfaction** | Overall satisfaction with ED visit | 4.54 ± 0.69 | 4.57 ± 0.73 | 0.45 |
| | Cleanliness | 4.51 ± 0.76 | 4.54 ± 0.74 | 0.36 |
| | Overall level of noise | 4.23 ± 0.85 | 4.26 ± 0.86 | 0.53 |
| | Respect to confidentiality and privacy | 4.55 ± 0.71 | 4.59 ± 0.67 | 0.20 |
| | Likelihood of recommending our ED to others | 4.68 ± 0.63 | 4.68 ± 0.63 | 0.93 |
| | Overall ED LOS | 4.24 ± 0.96 | 4.24 ± 0.98 | 0.99 |

difference in age between the two groups was not statistically significant for 9 out of the 11 months analyzed.

The ED followed recommendations from the EMR vendor and from other institutions that previously successfully implemented the EMR. Additional measures were taken, knowing that EMR implementation at our institution was the first in Lebanon and the area. The institution had to rely on its own staff and resources since no other institutions in the country or in nearby countries previously implemented this EMR, so a pool of experts and consultants was not available. Table 4 summarizes the vendor recommendations and also lists the measures that were followed in our low-resource setting.

Several EDs witnessed a prolonged patient LOS mainly in the immediate post-implementation period, but this effect was transient. LOS returned to baseline after a period of time that varied among institutions [14, 15, 20, 21]. Some other EDs, however, did not show a prolonged LOS after EMR implementation [2, 16]. An ED in Iowa reported a falsely elevated patient LOS after EMR implementation because patients underwent quick-registration before triage, whereas, prior to implementation, registration used to follow triage [22]. Similar to the

**Table 4. Vendor recommendations and specific recommendations for low-resource setting.**

| Category | Recommendations from Vendor and Other Institutions | Specific Additional Measures in Low-Resource Setting |
|---|---|---|
| Training | • Offering training classes as well as 'playground' for practice [5, 11, 14–16]<br>• Sending tips and updates to end-users after Go-Live [15] | • Initiating training early (4 months instead of 2 months prior to Go-Live)<br>• Early super-user training<br>• Using shadow charts shortly after Go-Live (documentation on both paper charts and EMR in order to ensure adequate documentation)<br>• Frequent simulation and drills with attending physicians' involvement |
| Super-users | • Having super-users usually from within the department [5, 11, 17] | • Selecting super-users from every staff category (attending physicians, residents, nurses, students, clerks. . .)<br>• Selecting super-users to cover poorly staffed shifts (such as night shifts)<br>• Super-users offering one-to-one training to end-users |
| Staffing | • Having a physician champion [8, 18, 19]<br>• Increasing staffing to compensate for slowness that might occur after Go-live [8, 15, 16] | • Having a dedicated cut-over team<br>• Having a higher super-user to staff ratio<br>• Increasing staffing in high turnover areas (low-acuity and urgent care) |
| Structural Readiness | • Installing adequate hardware including new computers, printers, and mobile workstations [5, 8, 15]<br>• Installing a dedicated workstation for every clinician [15] | • Establishing a multidisciplinary team to assess hardware needs and locations of stations<br>• Redesigning nursing station with predetermined assignments of workstations to different staff categories (physicians, nurses, etc. . .) |
| Workflows | • Creating order sets for common ED presentations [15] | • Assessing workflows in frequent meetings starting 2 years before Go-Live, especially high risk workflows (such as registration) and involving all stakeholders<br>• Identifying high risk areas early on in focus groups, and following up regularly on implementation and modification |
| Troubleshooting | • Developing an efficient troubleshooting process<br>• Having a downtime plan [5, 8] and a SWAT team | • Using online messaging systems where super-users discussed issues and suggested solutions |
| Staff Engagement | • Having frequent multidisciplinary meetings for optimization [18]<br>• Leadership commitment and buy-in [5, 8, 15, 19]<br>• Involvement of all ED staff including physicians, nurses, IT staff, administrative staff, and all other ED personnel [5] | • Having multidisciplinary teams that combine leaders and front-liners<br>• Having regular town hall meetings to demonstrate workflows as early as 3 months prior to implementation<br>• Creating steering and executive committees to oversee implementation at all stages<br>• Creating a clinical core owner group involved in end-user engagement to facilitate implementation across all risk areas |
| Reporting | • Creating and modifying the build for needed reports | • Focusing on reporting and early report validation plan (selecting key quality metrics and predefining the top quality metrics to be validated) |

experience of other EDs, our ED LOS increased in the post-implementation period, but returned to its baseline values after around 6 months despite adding a new interval post-implementation to LOS (door and quick-registration to triage). In the pre-implementation period, the patient was registered after triage.

Bounce-back rates slightly increased after EMR implementation. Several factors including seasonal, clinical, and patient-related might have contributed to this observation in our setting. In other institutions, these rates decreased [14] or remained unchanged [2].

Patient satisfaction is often cited to be negatively impacted by EMR implementation. Previous studies in EDs in the US reported that patient satisfaction was negatively affected by EMR implementation [14, 15]. Data from our ED showed that patient satisfaction metrics remained unchanged except for registration time before triage where a new step was introduced post implementation. Before implementation, patients used to go immediately to triage before registration. The post-implementation workflow required patients to complete a quick registration process prior to triage which could have negatively impacted satisfaction with the intake process.

This study has few limitations. Firstly, it was a retrospective study and therefore some data might be missing, especially in the pre-implementation period when data required manual cleaning. Secondly, patient satisfaction survey questions were kept the same despite EMR implementation. These questions remain valid since EMR implementation caused major

changes in ED workflows and new trends would have been detected in the survey results. Also, some patient characteristics (country, guarantor, and ESI) showed significant differences between the pre- and post-implementation groups even after the month-by-month analysis. We believe these differences may be due to the changes in the way of entering information in EMR. These differences are less likely to bias the results regarding the operational metrics such as LOS.

Various metrics were used to assess the impact of EMR implementation in this study. There are multiple other performance metrics that might have been affected by this EMR implementation, such as lab and radiology result turnaround time, financial metrics, and other operational metrics. However, this study was not intended to assess all aspects of implementation. The study used data collected as part of regular ED operations and monitoring. Data is validated on a regular basis and used for reporting and compared to benchmarks identified by Emergency Department Benchmarking Alliance (EDBA).

Different EDs have different processes and measuring time intervals can vary, but the proposed clinical metrics and operational metrics can be adopted in other settings to assess implementation.

## Conclusion

Lessons learned from this successful initiative include the importance of following strategies recommended by EMR vendor, previous approaches followed in other institutions as well as specific and tailored strategies used in our institution. All those strategies can be integrated into future implementation projects and used as part of the framework to ensure successful implementation in a low resource setting.

## Clinical relevance statement

EMR implementation can be successful in low-resource countries. In addition to the EMR vendor recommendations, additional strategies can be followed to ensure a successful EMR implementation. These include the presence of a dedicated multidisciplinary team to optimize workflows, a well-developed super-user strategy, and early initiation of training that includes simulations and drills.

1. Multiple choice questions

    1. What are some of the pre-implementation strategies adapted in our institution?

        A. ED leadership engagement

        B. Staff training and engagement

        C. Giving physician super-users clinical shifts

        D. Answer A & B

    The correct answer is D. ED leadership engagement as well as staff training and engagement were two of the pre-implementation strategies adopted in our institution. Having non-clinical shifts for super-users was a post-implementation strategy.

    ED leadership was engaged fully in all implementation phases. Extensive sessions over several months were dedicated to workflow analysis, ED processes, staffing and hardware requirements and preparation, and issues with interfaces. All clinical and non-clinical risks were reviewed, and corresponding mitigation plans were prepared. Paper-based order sets and protocols were modified and transferred to the EMR.

All ED physicians, nurses, and staff received online training, hands-on training, and access to the EMR 'playground' for practice. Super-users received advanced training mainly regarding troubleshooting in order to help colleagues during and after Go-Live. Each super-user was assigned a group of 4 to 5 trainees for one-to-one support as needed. Physicians who needed additional sessions were identified in advance and received additional personalized training from super-user physicians.

2. Super-users:

 A.  Included every staff category (attending physicians, residents, nurses, clerks)

 B.  Helped cover poorly staffed shifts

 C.  Offered one-to-one training to end-users

 D.  All of the above

The correct answer is D. Super-users, considered as advanced learners, were selected from within the ED, and they included attending physicians, residents, medical students, nurses, and other ED staff. They helped cover poorly staffed shift, such as night shifts. Super-users were assigned a group of 4 to 5 end-users for one-to-one training and support as needed.

## Acknowledgments

The authors acknowledge Layal Hamdar, Maha Makki, and Moustafa Al Hariri, PhD for their contributions to this article.

## Author Contributions

**Conceptualization:** Nagham Faris, Miriam Saliba, Hani Tamim, Mazen El Sayed, Eveline Hitti.

**Data curation:** Nagham Faris, Rima Jabbour, Ahmad Fakih, Zouhair Sadek, Rula Antoun, Mazen El Sayed, Eveline Hitti.

**Formal analysis:** Nagham Faris, Miriam Saliba, Hani Tamim, Mazen El Sayed, Eveline Hitti.

**Investigation:** Miriam Saliba, Rula Antoun, Mazen El Sayed, Eveline Hitti.

**Methodology:** Nagham Faris, Miriam Saliba, Rima Jabbour, Mazen El Sayed, Eveline Hitti.

**Project administration:** Nagham Faris, Rima Jabbour, Mazen El Sayed, Eveline Hitti.

**Resources:** Rula Antoun, Mazen El Sayed.

**Software:** Miriam Saliba, Rima Jabbour, Ahmad Fakih, Zouhair Sadek, Rula Antoun, Mazen El Sayed, Eveline Hitti.

**Supervision:** Nagham Faris, Mazen El Sayed, Eveline Hitti.

**Validation:** Hani Tamim, Rima Jabbour, Ahmad Fakih, Zouhair Sadek, Rula Antoun, Mazen El Sayed, Eveline Hitti.

**Visualization:** Miriam Saliba, Rima Jabbour, Ahmad Fakih, Zouhair Sadek, Rula Antoun, Mazen El Sayed, Eveline Hitti.

**Writing – original draft:** Nagham Faris, Miriam Saliba, Zouhair Sadek, Mazen El Sayed, Eveline Hitti.

**Writing – review & editing:** Nagham Faris, Miriam Saliba, Hani Tamim, Rima Jabbour, Ahmad Fakih, Mazen El Sayed, Eveline Hitti.

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
