## [Decision Letter · Decision Letter 0]

12 Dec 2022

PONE-D-22-31338Electronic Medical Record implementation in the Emergency Department in a low-resource country: Lessons learnedPLOS ONE

Dear Dr. Eveline Hitti,

Thank you for submitting your manuscript to PLOS ONE. After careful consideration, we feel that it has merit but does not fully meet PLOS ONE’s publication criteria as it currently stands. Therefore, we invite you to submit a revised version of the manuscript that addresses the points raised during the review process.

We look forward to receiving your revised manuscript.

Kind regards,

Aussama Khalaf Nassar, MD, MSc, FRCSC, FACS

Academic Editor

PLOS ONE

2. In the ethics statement in the manuscript and in the online submission form, please provide additional information about the patient records/samples used in your retrospective study. Specifically, please ensure that you have discussed whether all data/samples were fully anonymized before you accessed them and/or whether the IRB or ethics committee waived the requirement for informed consent. If patients provided informed written consent to have data/samples from their medical records used in research, please include this information.

Reviewers' comments:

Reviewer's Responses to Questions

**Comments to the Author**

1. Is the manuscript technically sound, and do the data support the conclusions?

Reviewer #1: Partly

Reviewer #2: Yes

Reviewer #3: Yes

2. Has the statistical analysis been performed appropriately and rigorously? 

Reviewer #1: I Don't Know

Reviewer #2: I Don't Know

Reviewer #3: Yes

3. Have the authors made all data underlying the findings in their manuscript fully available?

Reviewer #1: Yes

Reviewer #2: Yes

Reviewer #3: Yes

4. Is the manuscript presented in an intelligible fashion and written in standard English?

Reviewer #1: Yes

Reviewer #2: No

Reviewer #3: Yes

5. Review Comments to the Author

Reviewer #1: Although the patient characteristics are not a primary finding of the study and the focus is on the implementation of the system and its effect mainly on operational aspects and patient/staff satisfaction, I wonder why the patient populations are so different in the pre and post phases. This difference (in age for example), may have an effect on the medical complaints and diagnosis process, and bias the results about LOS. It is necessary to address this point in the manuscript. Assuming that the study includes all visits, it may be worthwhile to compare month by month and see if it is consistent, possibly omit months with such differences and shorten the study period so patients are more similar. Regarding the Lebanese/other classification - it may be that missing data in the pre phase were classified as "other" resulting in the distribution appearing in table 1. The paper would be much stronger (and valid) if pre-post populations were more similar, otherwise you can not deduce that the changes (or lack of) are a result of the new EMR system.

The paper needs a deeper description/discussion on what it means to be a "low resource country" (where and why does this make a difference in comparison to other countries).

Reviewer #2: Hello.

The authors had summarized the main research question and key findings.

The authors had discussed the limitations of the study.

The manuscript text supports the data shown in the tables.

I recommend language editing.

Reviewer #3: I would like to congratulate author and the team on the effort provided in the study. The study provided good insight on the outcome of digital health system implementation in a low resource country.

6. PLOS authors have the option to publish the peer review history of their article (what does this mean?). If published, this will include your full peer review and any attached files.

Reviewer #1: No

Reviewer #2: No

Reviewer #3: **Yes: **Nurul Huda Ahmad

---

## [Author Response · Author response to Decision Letter 0]

17 Feb 2023

January 26, 2023

Subject: point-by-point reply to comments

Re: PONE-D-22-31338

Dear Reviewers,

Thank you for your consideration of our Manuscript PONE-D-22-31338 entitled " Electronic Medical Record implementation in the Emergency Department in a low-resource country: Lessons learned". 

Please find below a point-by-point reply to your comments. Comments will be copied verbatim, followed immediately by our response. Should we need to quote a section from the manuscript, it will be done between quotation marks.

Reviewer #1: Although the patient characteristics are not a primary finding of the study and the focus is on the implementation of the system and its effect mainly on operational aspects and patient/staff satisfaction, I wonder why the patient populations are so different in the pre and post phases. This difference (in age for example) may have an effect on the medical complaints and diagnosis process, and bias the results about LOS. It is necessary to address this point in the manuscript. Assuming that the study includes all visits, it may be worthwhile to compare month by month and see if it is consistent, possibly omit months with such differences and shorten the study period so patients are more similar. Regarding the Lebanese/other classification - it may be that missing data in the pre phase were classified as "other" resulting in the distribution appearing in table 1. The paper would be much stronger (and valid) if pre-post populations were more similar, otherwise you can not deduce that the changes (or lack of) are a result of the new EMR system.

The paper needs a deeper description/discussion on what it means to be a "low resource country" (where and why does this make a difference in comparison to other countries).

Due to the large sample size, the differences between the patient characteristics between the pre- and post-implementation groups are statistically significant but not necessarily clinically significant. A month-by-month analysis was done and is summarized in the table in Appendix 1 of this document. This analysis was not added to the manuscript, but a summary of the findings was added.

We added the following to our discussion: “Statistically significant differences are seen in the demographic characteristics between pre and post groups. However, most of these differences are mainly due to the large sample size and are not clinically significant. The pre-implementation patient group seems to be significantly older than the post-implementation group, which might seem to bias the operational metrics results such as LOS. For this reason, a month-by-month analysis was done, and the difference in age between the two groups was not statistically significant for 9 out of the 11 months analyzed.”

The following was added to the limitations: “Some patient characteristics (country, guarantor, and ESI) showed significant differences between the pre- and post-implementation groups even after the month-by-month analysis. We believe these differences may be due to the changes in the way of entering information in EMR. These differences are less likely to bias the results regarding the operational metrics such as LOS.”

In the methods section, a description of the low-resource setting was added.

“This EMR implementation was the first of its kind in Lebanon, a low-middle income country. In this low-resource setting, financial resources were limited. There is an absence of government funding for healthcare infrastructure, and healthcare institutions rely on internal funding. Moreover, there was a lack of trained personnel to implement and maintain the EMR. Budget to hire external consultants and to pay faculty and staff additional compensation for roles performed during Go-Live was limited.”

Reviewer#2: Hello.

The authors had summarized the main research question and key findings.

The authors had discussed the limitations of the study.

The manuscript text supports the data shown in the tables.

I recommend language editing.

The manuscript was revised and language errors were corrected.

Reviewer #3: I would like to congratulate author and the team on the effort provided in the study. The study provided good insight on the outcome of digital health system implementation in a low resource country.

Thank you for your comments.

 

**Appendix 1: Table is provided in the uploaded file under "response to reviewers."

---

## [Decision Letter · Decision Letter 1]

17 Jan 2024

Electronic Medical Record implementation in the Emergency Department in a low-resource country: Lessons learned

PONE-D-22-31338R1

Dear Dr. Hitti,

We’re pleased to inform you that your manuscript has been judged scientifically suitable for publication and will be formally accepted for publication once it meets all outstanding technical requirements.

Kind regards,

Asli Suner Karakulah, PhD

Academic Editor

PLOS ONE

Additional Editor Comments (optional):

Reviewers' comments:

Reviewer's Responses to Questions

**Comments to the Author**

1. If the authors have adequately addressed your comments raised in a previous round of review and you feel that this manuscript is now acceptable for publication, you may indicate that here to bypass the “Comments to the Author” section, enter your conflict of interest statement in the “Confidential to Editor” section, and submit your "Accept" recommendation.

Reviewer #3: All comments have been addressed

Reviewer #4: All comments have been addressed

2. Is the manuscript technically sound, and do the data support the conclusions?

Reviewer #3: Yes

Reviewer #4: Yes

3. Has the statistical analysis been performed appropriately and rigorously? 

Reviewer #3: Yes

Reviewer #4: Yes

4. Have the authors made all data underlying the findings in their manuscript fully available?

Reviewer #3: Yes

Reviewer #4: Yes

5. Is the manuscript presented in an intelligible fashion and written in standard English?

Reviewer #3: Yes

Reviewer #4: Yes

6. Review Comments to the Author

Reviewer #3: (No Response)

Reviewer #4: (No Response)

7. PLOS authors have the option to publish the peer review history of their article (what does this mean?). If published, this will include your full peer review and any attached files.

Reviewer #3: **Yes: **Nurul Huda Ahmad

Reviewer #4: No

---

## [Editor Report · Acceptance letter]

20 Feb 2024

PONE-D-22-31338R1 

PLOS ONE

Dear Dr. Hitti, 

I'm pleased to inform you that your manuscript has been deemed suitable for publication in PLOS ONE. Congratulations! Your manuscript is now being handed over to our production team.

Kind regards, 

on behalf of

Dr. Asli Suner Karakulah 

Academic Editor

PLOS ONE